# Choosing the right basis for interpretability: Psychophysical comparison between neuron-based and dictionary-based representations

**Julien Colin**
*ELLIS Alicante, Alicante, Spain*
*Carney Institute for Brain Science, Brown University, USA*

*julien@ellisalicante.org*

**Lore Goetschalckx**
*IMEC, Leuven, Belgium*

*lore.goetschalckx@imec.be*

**Thomas Fel**
*Kempner Institute, Harvard University, USA*

*tfel@fas.harvard.edu*

**Victor Boutin**
*CerCo - CNRS, University of Toulouse, France*

*victor_boutin@brown.edu*

**Thomas Serre**[*]
*Carney Institute for Brain Science, Brown University, USA*

*thomas_serre@brown.edu*

**Nuria Oliver**[*]
*ELLIS Alicante, Alicante, Spain*

*nuria@ellisalicante.org*

## Abstract

Interpretability research often adopts a neuron-centric lens, treating individual neurons as the fundamental units of explanation. However, neuron-level explanations can be undermined by superposition, where single units respond to mixtures of unrelated patterns. Dictionary learning methods, such as sparse autoencoders and non-negative matrix factorization, offer a promising alternative by learning a new basis over layer activations. Despite this promise, direct human evaluations comparing neuron-based and dictionary-based representations remain limited.

We conducted three large-scale online psychophysics experiments (N=481) comparing explanations derived from neuron-based and dictionary-based representations in two convolutional neural networks (ResNet50, VGG16). We operationalize interpretability via visual coherence: a basis is more interpretable if humans can reliably recognize a common visual pattern in its maximally activating images and generalize that pattern to new images. Across experiments, dictionary-based representations were consistently more interpretable than neuron-based representations, with the advantage increasing in deeper layers.

Critically, because models differ in how neuron-aligned their representations are—with ResNet50 exhibiting less dictionary-axis alignment, consistent with more distributed and potentially more superposed representations, neuron-based evaluations can mask cross-model differences, such that ResNet50's higher interpretability emerges only under dictionary-based comparisons.

These results provide psychophysical evidence that dictionary-based representations offer a stronger foundation for interpretability and caution against model comparisons based solely on neuron-level analyses.

---

[*]co-principal investigators

# 1 Introduction

A central goal of explainable AI (XAI) in computer vision is to identify the visual elements that drive the decisions of deep neural networks (DNNs) (Selvaraju et al., 2017; Bau et al., 2017; Kim et al., 2018; Ghorbani et al., 2019; Cammarata et al., 2020b; Morcos et al., 2018). Doing so requires recovering the visual patterns that systematically influence internal activations and ultimately shape model outputs.

This goal is shared with the study of biological vision, where decades of work have sought the "preferred stimulus" of individual neurons in the visual cortex (Hubel & Wiesel, 1959; Quiroga et al., 2005). Early neuroscience-inspired XAI similarly focused on neuron-level analyses (Zhou et al., 2016; Bau et al., 2017), alongside methods that synthesize maximally activating images for individual units (e.g., Erhan et al., 2009; Zeiler & Fergus, 2014; Olah et al., 2017).

In this context and for the remainder of this paper, we use the term *neuron* to refer to the unit of a layer: an individual neuron in fully-connected ReLU layers and a channel (feature map) in convolutional layers.

At the same time, neuroscience has increasingly shifted from single-neuron selectivity to population codes (see Ebitz & Hayden (2021)), reflecting the view that neural representations are sparse and distributed rather than purely local (Haxby et al., 2001; Quiroga et al., 2008). A similar shift is emerging in XAI because neuron axes can be a poor explanatory basis under the "superposition" hypothesis (Elhage et al., 2022; Fel et al., 2023a): when a model encodes more patterns than it has units, individual neurons may respond to mixtures of unrelated patterns—often referred to as polysemantic neurons.

To address this challenge, recent work applies dictionary learning methods (Fel et al., 2023b; Bricken et al., 2023; Gao et al., 2024; Costa et al., 2025; Fel et al., 2025; Zaigrajew et al., 2025; Bussmann et al., 2025) to learn a dictionary basis over layer activations, ideally yielding directions that correspond to single, human-recognizable patterns. From an interpretability perspective, a learned dictionary basis is appealing because it aims to replace those polysemantic neuron axes with more monosemantic directions (Fig. 1).

Despite growing interest in dictionary-based representations, direct human evidence that they provide a better foundation for interpretability than neuron-based representations remains limited. This paper aims to fill this gap. Furthermore, model comparisons remain underexplored and have largely relied on neuron-based evaluations (Zimmermann et al., 2023; 2024); if models differ in their degree of superposition, conclusions drawn from neuron-based analyses may be unreliable.

We operationalize interpretability via *visual coherence*: a basis is more interpretable to the extent that humans can reliably identify the common visual pattern in its maximally activating images and generalize that pattern to new images[1]. We additionally require that the directions evaluated within a basis be demonstrably involved in the model's decisions.

We conducted three large-scale psychophysics experiments (N=481 participants, 16,835 responses) comparing the interpretability of neuron-based and dictionary-based representations in ResNet50 and VGG16. Our main contributions are:

- We identify a semantic confound in widely used human-evaluation protocols (Borowski et al., 2021; Zimmermann et al., 2023) and introduce a control that mitigates it.

- Across experiments, we find that dictionary-based representations are consistently more interpretable than neuron-based representations, with the advantage increasing in deeper layers.

- By quantifying axis-alignment[2], we find that ResNet50 exhibits lower axis-alignment of its dictionary elements than VGG16, indicating more distributed representations which is consistent with a greater degree of superposition than VGG16, obscuring cross-model differences in neuron-based analyses that become apparent only in dictionary-based comparisons, revealing ResNet50 to be the more interpretable model.

---

[1]In the remainder of this paper, we use *interpretability* to refer specifically to this notion of human-recognizable visual coherence.

[2]Through the paper, axis-alignment refers to the alignment of the representation with neuron axes.

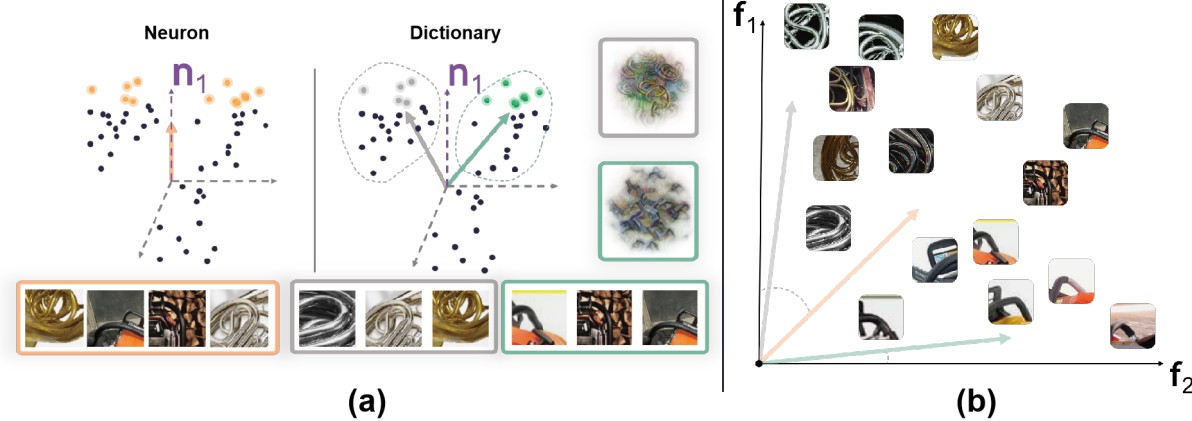

Figure 1: **(a)** ● **Neuron (axis) basis** versus ● **Dictionary basis**. Individual neurons may respond to multiple unrelated visual patterns (bottom), whereas dictionary learning aims to recover feature directions that isolate simpler patterns. **(b)** Dictionary learning yields a new basis over activations whose elements ideally correspond to single patterns. The interpretability hope is that these patterns align with the set of human-recognizable concepts $S = \{f_1, f_2, ..., f_n\}$.

- We empirically demonstrate that the choice of representational basis materially affects model comparisons, cautioning against drawing conclusions from neuron-level analyses alone.

## 2 Related work

**From neuron-based to dictionary-based representations.** Early work in explainable AI (XAI) for computer vision developed attribution methods (e.g., Zeiler & Fergus, 2014; Sundararajan et al., 2017; Selvaraju et al., 2017) to explain individual predictions. These approaches primarily address the *where* question, that is, identifying which pixels or regions are most influential for a given output. However, they often fall short on the *what* question (Kim et al., 2018; Colin et al., 2022): the visual patterns or factors the model relies on to make its decision. This limitation motivated the development of feature visualization methods (Nguyen et al., 2016; Olah et al., 2017), which aim to characterize what neurons or layers represent. A consistent finding is that single neurons can respond to multiple, visually distinct patterns (Nguyen et al., 2016; Cammarata et al., 2020a; Bricken et al., 2023), echoing similar observations in natural language processing (Elhage et al., 2022).

These polysemantic responses suggest that neuron axes may not align with the underlying factors of variation in a model's representations. Under the superposition hypothesis (Elhage et al., 2022; Fel et al., 2023a), models can encode more patterns than they have neurons, causing individual units to act as mixtures of multiple patterns. In that regime, interpreting single neurons may be no more principled than interpreting arbitrary directions in the activation space, motivating the search for alternative bases that better isolate interpretable elements. This perspective has spurred increased interest in learning dictionary bases over latent activations via dictionary learning and related concept-extraction methods (Ghorbani et al., 2019; Fel et al., 2023b). In parallel, sparse autoencoders (SAEs) have emerged as a promising approach for discovering more monosemantic directions in deep networks (Bricken et al., 2023; Cunningham et al., 2023; Gao et al., 2024; Costa et al., 2025; Fel et al., 2025; Zaigrajew et al., 2025; Bussmann et al., 2025). In this paper, we compare the suitability of the neuron basis versus a learned dictionary basis as a foundation for interpretability in computer vision models.

**Human-evaluation of interpretability.** Since a primary goal of XAI is to make model behavior understandable to people, interpretability ultimately requires human-centered evaluation. Psychophysics-style experiments provide a direct benchmark for assessing whether an explanation is comprehensible and usable to human observers, complementing automated or proxy metrics.

Borowski et al. (2021) were the first to quantify the interpretability of deep neural network representations using psychophysics experiments. Their protocol visualized unit selectivity by contrasting maximally and minimally activating natural stimuli. In particular, they evaluated neuron-based representations (single-unit activations) in an Inception V1 (Szegedy et al., 2015) trained on ImageNet (Deng et al., 2009). In each trial, participants viewed sets of maximally and minimally activating images for a given neuron and then selected which of two query images also strongly activated that neuron (see Fig. 2). They concluded that, from a human-centered perspective, natural exemplars are more effective than synthetically generated feature visualizations (Olah et al., 2017).

Zimmermann et al. (2021) proposed a variation of this task to investigate if humans gained causal insights from those visualizations. Participants predicted the effect of an intervention (*e.g.*, occluding an image region) on unit activation. They found that synthetic feature visualizations could improve performance, but provided limited advantage over natural exemplar images.

Interestingly, in a similar vein but in NLP, Bricken et al. (2023) compared the interpretability of 162 elements drawn from both neuron-based and dictionary-based representations. In their study, a single author rated each element using examples sampled across its activation range. They reported that dictionary-based elements were substantially more interpretable than individual neurons. Our work differs in domain (vision rather than language), experimental design, and scale (a single rater versus 16,835 behavioral responses from 481 participants).

**Comparative analyses of model interpretability.**    Zimmermann et al. (2023) were the first to compare the interpretability of models using human evaluations. They extended the work of Borowski et al. (2021) to a broader range of computer vision architectures, including ResNet50 (He et al., 2016), and concluded that increasing model scale does not enhance interpretability. While we both build on the experimental protocol introduced by Borowski et al. (2021), their focus is on scaling these original insights across architectures. In contrast, we adapt the same protocol to address a different research question: which type of representation (neuron-based or dictionary-based) is more interpretable to humans. To the best of our knowledge, this is the first study that compares models using dictionary-based representations.

To further scale the evaluation of model interpretability, Zimmermann et al. (2024) automated the human evaluation protocol introduced by Borowski et al. (2021). In each trial, human judgments were approximated by computing pairwise perceptual similarities between queries and explanations, which were then used by a binary classifier to predict the correct query. The resulting metric was found to correlate strongly with previous results (Zimmermann et al., 2023). This work shows that although most models contain a substantial number of interpretable units, differences across models primarily arise from the prevalence of highly uninterpretable units. Such units are often attributed to superposition, whose impact may vary across models. Consequently, comparing models solely based on neuron-based representations may disproportionately penalize certain architectures, potentially yielding misleading conclusions. Our work compares models using both neuron-based and dictionary-based representations and demonstrates that the choice of representational basis can substantially impact comparative outcomes.

## 3   Methodology

In this section, we first provide an overview of the technical methods used in our experiments (named Experiments I, II and III), followed by a description of the psychophysical experiments conducted to compare the interpretability of neuron-based versus dictionary-based representations.

### 3.1   Technical methods

**Models**   Experiments I and II described below used a ResNet50 (He et al., 2016) and Experiment III used a VGG16 (Simonyan & Zisserman, 2014), both sourced from the Torchvision (Marcel & Rodriguez, 2010) library and pre-trained on ImageNet-1k (Deng et al., 2009). We focus on convolutional neural networks (CNNs) because they remain widely used in practical computer vision applications, making our findings broadly relevant, and they produce the positive activations required by our dictionary learning method.

**Dictionary-based representations.** To compute dictionary-based representations for the psychophysics experiments (see Section 3.2), we employed CRAFT (Fel et al., 2023b), a dictionary learning method based on Non-negative Matrix Factorization (NMF). Specifically, given a model $f : \mathcal{X} \to \mathcal{A}$ that maps from an input space $\mathcal{X} \subseteq \mathbb{R}^d$ to an activation space $\mathcal{A} \subseteq \mathbb{R}^p$ (*i.e.,* any layer of the network), we compute the activations $\mathbf{A} = f(\mathbf{X}) \in \mathbb{R}^{n \times p}$, where $\mathbf{X} = [\mathbf{x}_1, \ldots, \mathbf{x}_n]^\top \in \mathbb{R}^{n \times d}$ represents a set of $n$ input data points. Each row $\boldsymbol{a}_i \geq \mathbf{0}$ of $\mathbf{A}$ contains the non-negative activations for a given data point $\mathbf{x}_i$, due to the use of ReLU activations. NMF approximates $\mathbf{A}$ as:

$$(\mathbf{Z}^\star, \mathbf{D}^\star) = \underset{\mathbf{Z} \geq \mathbf{0}, \ \mathbf{D} \geq \mathbf{0}}{\arg\min} \left\| \mathbf{A} - \mathbf{Z}\mathbf{D}^\top \right\|_F,$$

where $\|\cdot\|_F$ denotes the Frobenius norm. Here, $\mathbf{Z} \in \mathbb{R}^{n \times k}$ are the **codes**, and $\mathbf{D} \in \mathbb{R}^{p \times k}$ forms the **dictionary** with k elements. Both $\mathbf{Z}$ and $\mathbf{D}$ are constrained to have non-negative entries and tend to be sparse due to the properties of NMF. The dictionary matrix $\mathbf{D}$ provides a new set of basis vectors aligned with the activation patterns of the neural network, while $\mathbf{Z}$ contains the coefficients representing the original activations $\mathbf{A}$ in terms of these basis vectors.

We use NMF rather than sparse autoencoders (SAEs) because NMF respects the geometry of post-ReLU activations: both the dictionary and codes are constrained to be non-negative, whereas SAEs can learn negative dictionary atoms corresponding to directions that cannot exist in the activation space (Fel et al., 2023a). NMF is also more stable, avoiding the hyperparameter sensitivity inherent in SAE training (Fel et al., 2025).

**Element importance.** Let $\mathbf{v}$ be a vector from an intermediate representation, either a dictionary element in $\mathbf{D}$ or a neuron axis in $\mathcal{A}$. Let $\mathbf{x}$ be an input that strongly activates $\mathbf{v}$, and let $f(\mathbf{x})$ denote the corresponding logit score. To assess the importance of $\mathbf{v}$, we measure how sensitive $f(\mathbf{x})$ is to perturbations of $\mathbf{v}$ using Gradient×Input (Shrikumar et al., 2017), which has been shown to provide a faithful importance measure in latent spaces (Fel et al., 2023a):

$$\mathrm{GI}(\mathbf{x}, \mathbf{v}) \ = \ \mathbf{v} \odot \frac{\partial f(\mathbf{x})}{\partial \mathbf{v}}. \tag{1}$$

The more important $\mathbf{v}$ is for the model's decision at $\mathbf{x}$, the larger $\mathrm{GI}(\mathbf{x}, \mathbf{v})$.

**Axis-alignment of dictionary elements.** If a model contains polysemantic neurons, then a learned dictionary is expected to recover directions that are not well-approximated by any single neuron axis. To quantify how closely each dictionary element aligns with the neuron (coordinate) basis—or equivalently, how distributed it is across neurons—we use the sparsity measure of Hoyer (2004). We note that axis-alignment measures the structure of the dictionary matrix $\mathbf{D}$—specifically, how concentrated each basis vector is on individual neuron axes—and should not be conflated with the sparsity of the codes $\mathbf{Z}$ or with the orthogonality of the representation. In the non-negative setting enforced by NMF and ReLU, lower axis-alignment indicates that a recovered feature direction is distributed across multiple neurons rather than concentrated on a single neuron. Such distributed representations are often associated with superposition, but axis-alignment does not provide a direct measure of the number of encoded features or de degree to which those features overlap.

Let $D \in \mathbb{R}^{p \times k}$ denote the learned dictionary for a layer with $p$ neurons, and let $\mathbf{d}_\ell = D_{:,\ell} \in \mathbb{R}^p$ be the $\ell$-th dictionary element expressed in the neuron basis. We define the average axis-alignment score as

$$\mathcal{H}(D) \ = \ \frac{1}{k} \sum_{\ell=1}^{k} \frac{\sqrt{p} - \|\mathbf{d}_\ell\|_1 / \|\mathbf{d}_\ell\|_2}{\sqrt{p} - 1}. \tag{2}$$

Here $\|\cdot\|_1$ and $\|\cdot\|_2$ denote the $\ell_1$ and $\ell_2$ norms. For nonnegative vectors, this index lies in $[0, 1]$: it is 0 for maximally dense vectors (mass evenly spread across coordinates) and approaches 1 for one-hot vectors (mass concentrated on a single neuron axis). Thus, higher values indicate that dictionary elements are closer to individual neuron axes, *i.e.,* more coordinate-sparse or axis-aligned, whereas lower values indicate elements that combine many neuron axes.

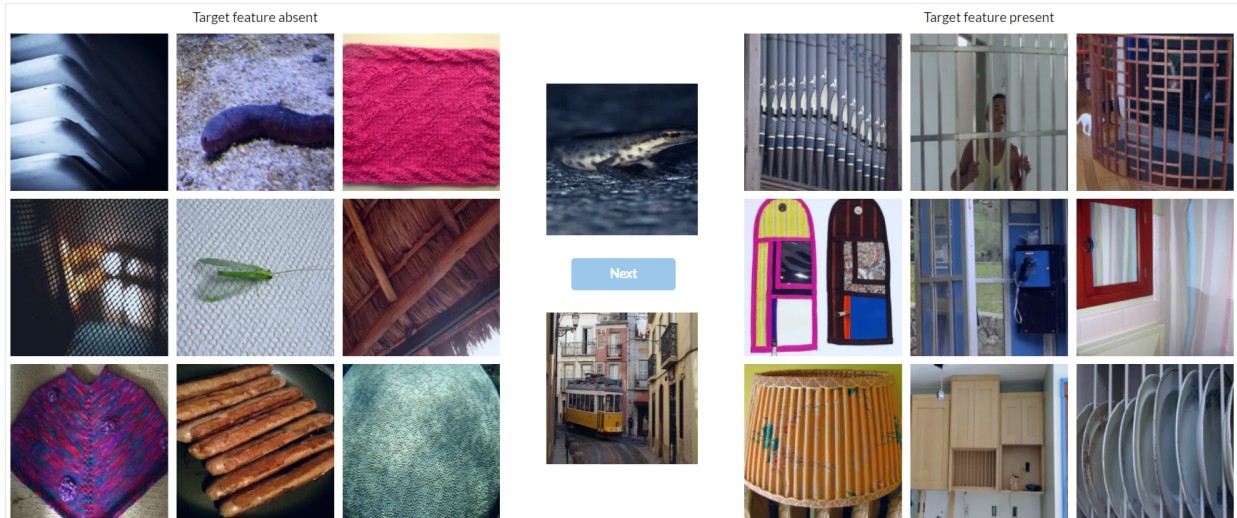

Figure 2: **Illustration of a trial.** Example of a trial in our study corresponding to Experiment I, dictionary-based condition for a unit located in $layer2.0.bn2$. Two panels of 9 reference images are located on the left and right-hand side of the display, separated by 2 query images in the center. Participants were asked to select the query image they believed shared the same visual pattern as the reference images displayed on the right panel, corresponding to maximally activating stimuli. The less ambiguous this shared pattern is, the more visually coherent the set of images and the more likely participants are to select the correct query. In this case, the correct query is the bottom image depicting a yellow tram.

## 3.2 Psychophysics experiments

### 3.2.1 Experimental protocol

Interpretability is ultimately a human-centered concept. To examine how the choice of representational basis shapes human judgments, we conducted three large-scale online psychophysics[3] experiments comparing units expressed either in the neuron basis or in a learned dictionary basis. Throughout, we use *visual pattern* (or *visual feature* in the perceptual sense) to refer to recurring, human-recognizable regularities in images, and we reserve *dictionary element* for a learned direction (dictionary element) in the activation space. Across all experiments, we evaluate how easily observers can identify the *consistent visual pattern* associated with a unit (neuron axis or dictionary element) from its maximally activating stimuli.

Because standard interpretability evaluations do not easily scale to large numbers of units (Colin et al., 2022), we adapted the psychophysics protocol of Borowski et al. (2021) to measure a unit's *visual coherence*, or equivalently, its lack of ambiguity, as a proxy for interpretability.

Each participant was assigned to one of two between-subject conditions: the neuron-basis condition or the dictionary-basis condition. After completing a practice session of 9 trials, participants performed 40 trials of the same task, with each trial corresponding to a different unit (a specific neuron axis or a dictionary element, depending on the condition).

In each trial, participants were shown two panels of 9 reference images, one on each side of the screen, separated by two *query* images in the center (Fig. 2). The right panel displayed images selected to strongly activate the target unit (maximally activating stimuli). The left panel displayed images intended to contrast with the right panel (the exact selection procedure depends on the experiment; see Section 3.2.3). Participants

---

[3]In this paper, we use the term *psychophysics* to refer to the broader area of developing rigorous human behavioral paradigms that quantitatively measure perception, beyond classical threshold or detection paradigms. This usage of the term psychophysics is aligned with that of Borowski et al. (Borowski et al., 2021) on whose protocol we build.

were asked to select the query image that they believed matched the reference images on the right, *i.e.*, the query image that shared the same consistent visual pattern as the maximally activating reference set.

Intuitively, when the maximally activating images are visually coherent, participants should more reliably select the correct query. Visual coherence suggests the unit's activation is driven by a consistent, recognizable pattern. Conversely, if the maximally activating images are heterogeneous (consistent with polysemanticity), the task becomes more ambiguous and accuracy should drop. We therefore summarize performance as the proportion of correct responses per unit, which serves as our primary measure of visual coherence and hence interpretability.

### 3.2.2 Unit and stimuli selection

**Unit selection.** Following Zimmermann et al. (2023), we construct a representative sample of units across different layers of the neural networks of interest by adopting their proposed sampling procedure. Specifically, we sampled neurons by first selecting a network layer uniformly at random from the set of layers of interest, and then selected a neuron uniformly at random from within the chosen layer. This two-stage sampling scheme was used instead of sampling uniformly over all neurons to avoid bias toward later layers, which typically contain more units in CNNs. We repeated this procedure to obtain 80 units in total, spanning multiple layers of the network (Zimmermann et al., 2023). Because CRAFT requires non-negative activations (Fel et al., 2023b), we mapped neurons from convolutional layers to their counterparts in the subsequent batch normalization layer (*e.g.*, $layer1.1.conv1$ neuron 53 -> $layer1.1.bn1$ neuron 53), which are followed by ReLU and thus yield non-negative post-ReLU activations. For VGG16 (Simonyan & Zisserman, 2014), we randomly selected 80 neurons across the ReLU layers. In total, we evaluated 80 units per model, distributed across 43 layers for ResNet50 and 11 layers for VGG16.

**Stimuli selection for the neuron-based condition.** For each of the selected neurons, we identified a representative set of 2,900 images from the validation set of ImageNet ILSVRC 2012 (Russakovsky et al., 2015): the 2,500 most strongly activating images and the 400 least strongly activating images. Following Borowski et al. (2021), we illustrated each neuron's selectivity through both maximally activating stimuli (images likely to contain the relevant visual pattern; see Fig.2, right panel) and minimally activating stimuli (images unlikely to contain it; see Fig.2, left panel). For the maximally activating panel, we selected a random sample of 9 images from the top 150; for the minimally activating panel, we uniformly sampled 9 images from the bottom 20. We created 10 different trials per neuron following this procedure to ensure image independence across results.

**Stimuli selection for the dictionary-based condition.** Our hypothesis is that applying dictionary learning to neuron-based representations allows us to recover alternative directions capturing the distributed structure that is not aligned with individual neurons. Such structure is often associated with superposition and may provide a more interpretable basis for analysis than the original neuron basis. To test this, we derived a complementary dictionary-based representation for each neuron in the neuron-based condition. Specifically, for each neuron in the neuron-based condition, we selected its top N=300 maximally activating images, *i.e.*, those that most strongly activated it. We then applied CRAFT to this set of images to learn a dictionary of k=10 elements over the activations of that neuron's layer (we select $N$ and $k$ following the recommendations in Fel et al. (2023b)). From these 10 dictionary elements, we selected the one that was most frequently maximally activated across the 300 images (see Appendix A for more details). Finally, we ranked the 2,900 images according to their activation along the chosen dictionary element to obtain the set of stimuli that illustrate the corresponding visual pattern.

### 3.2.3 Experiments

In this section, we describe each of the three psychophysics experiments conducted in this work. Each experiment consists of 1,600 trials (80 units × 2 conditions × 10 trials per unit).

**Experiment I.** This experiment is an adaptation of the methodology proposed by Borowski et al. (2021), conducted on a ResNet-50, with two conditions: neuron-based versus dictionary-based representations. An

illustration of a trial from Experiment I can be found in Fig. 2, and the experimental protocol is described in Section 3.2.1.

**Experiment II.**   The main objective of Experiment II was to control for a potential semantic confound uncovered in Experiment I. If the reference images that contain or do not contain the pattern of interest belong to distinct semantic categories, then it may be possible to solve the task through simple semantic grouping. Fig. A1 illustrates this phenomenon: in the depicted example, it is easier to solve the trial by inferring that the pattern of interest is *not* about a monkey than to identify the actual visual pattern present in the reference images. In such cases, correctly solving the trial tells us little about the interpretability of the unit. To mitigate this semantic confound, we proceeded as follows. Given a set of reference images containing the pattern of interest, we extracted their semantic labels from ImageNet and searched the 400 minimally activating images for a set of 9 images matching these semantic labels. When ImageNet labels were insufficient, we expanded the search by moving upward through the WordNet (Fellbaum, 2010) hierarchy. After up to 4 iterations, all but one unit were successfully controlled. This unit was excluded from both the neuron-based and dictionary-based conditions.

**Experiment III.**   To compare the interpretability across models, Experiment III applied the protocol from Experiment II to a VGG16.

### 3.2.4   Participants

A total of 481 participants were recruited for Experiments I, II, and III through the Prolific [4] online platform. All participants were native English speakers who reported no visual impairments and completed the study on a laptop or desktop computer (not mobile devices). They provided informed consent electronically and were compensated \$2.75 for their time, corresponding to \$15 USD per hour (approximately 10–13 minutes). The protocol was approved by the Institutional Review Board (IRB) of an institution affiliated with the authors. Based on the power analysis of Zimmermann et al. (2023), a minimum of 60 participants per condition (*i.e.*, 120 participants per experiment) was needed to obtain statistically robust results ($p < 0.05$). This prior was used given the closely matched experimental design. A sensitivity analysis in Appendix D further shows that our retained sample sizes were adequate to detect medium-sized dictionary-vs-neuron effects. Participants were required to: (1) succeed in at least 5 of the 9 practice trials, (2) correctly answer at least 4 of the 5 catch trials (attentiveness tests) randomly inserted throughout the experiment, and (3) complete the experiment within 3 standard deviations of the mean completion time for that experiment.

**Trial assignment.**   Each participant was assigned to a single condition (neuron-level or dictionary-level), making `Representation` a between-subject factor. For each participant, we (i) randomly sampled 40 of the 80 units and (ii) for each sampled unit, randomly drew 1 of its 10 trials. Participants therefore performed 40 trials, each corresponding to a different unit. No participant saw the same unit twice. Each experiment comprised 80 units x 2 conditions x 10 trials per unit = 1,600 unique trials. The 10 independent trials per unit (drawn from different random image samples within the top-150 pool) were distributed across different participants, such that each trial was seen by an approximately equal number of participants. `Depth` and, across experiments, `Model` were within-subject factors, as each participant saw units from multiple layers of one model.

## 4   Results

### 4.1   Dictionary elements are at least as decision-relevant as individual neurons

Interpretability aims to enable humans to understand the decision-making processes of machine learning models. Accordingly, it is essential to first verify that the elements under investigation genuinely influence model decisions. We therefore assess the relative importance of dictionary elements relative to neuron axes, confirming that the former are at least as informative as the latter for driving model outputs.

---

[4]`www.prolific.com`

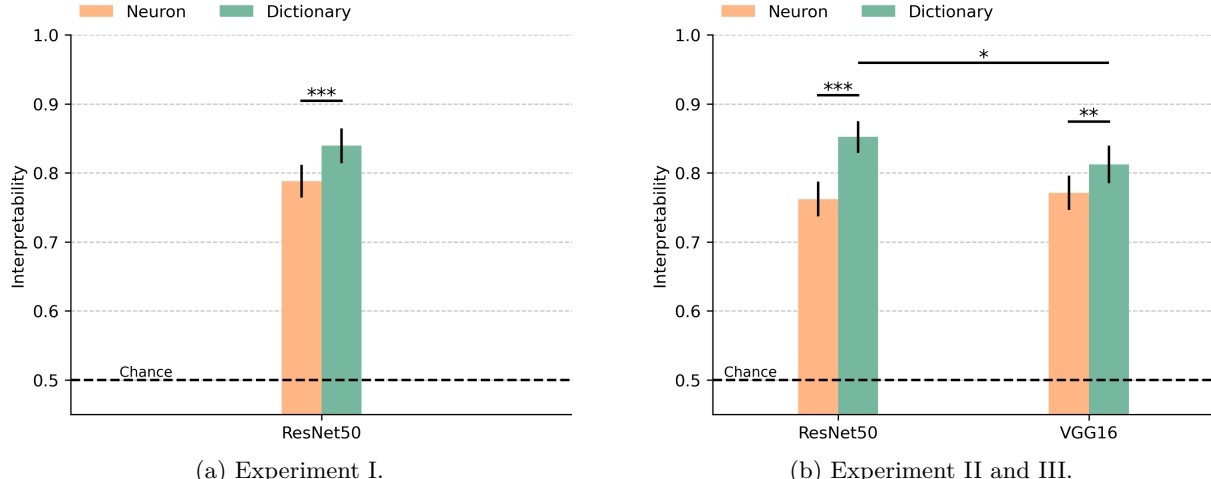

(a) Experiment I.   (b) Experiment II and III.

Figure 3: **Results for (a) Experiment I, and (b) Experiments II (ResNet50) and III (VGG16).**
Given a unit and a set of images illustrating it, we assess how visually coherent participants find this set of images, or equivalently, how unambiguous the underlying pattern is. Specifically, we measure the proportion of trials in which participants correctly identified the query image belonging to this set of images and refer to this value as *Interpretability*. Across all experiments, participants found it significantly easier to identify the consistent pattern in the dictionary-based condition than in the neuron-based condition. Additionally, ResNet50 is more interpretable than VGG16 in the dictionary-based condition but not in the neuron-based condition.

We quantify the importance of each element used in our psychophysics experiments by measuring the sensitivity of model decisions to perturbations of that element across its 300 most activating images (see Eq. 1). For each matched pair, we compute the difference in importance between the dictionary element and its corresponding neuron. We find no significant difference in ResNet50 ($M = 0.0002, SD = 0.0018$), $t(77) = 0.93$, $p = 0.36$, while dictionary elements are slightly more important than neurons in VGG16 ($M = 0.002, SD = 0.005$), $t(76) = 3.34$, $p = 0.001$ (One-sample t-test against 0).

These results show that dictionary elements have similar influence on model outputs as their corresponding neurons, attesting to the relevance of the selected elements for interpretability and supporting the fairness of our comparisons by matching elements of comparable importance.

## 4.2   Interpretable features should be understandable by humans

In this section, we summarize the results obtained from analyzing the responses from 133, 130, and 129 participants who successfully completed Experiments I, II, and III.

**Replication of previous findings.**   The experimental protocol employed in Experiment I for the neuron-based condition is the same as that described by Zimmermann et al. (2023). Thus, we first assess the extent to which our results replicate previous findings. For ResNet50, Zimmermann et al. (2023) report an average task performance of $83.0\% \pm 2.0$ [5]. In our experiment, we obtain an average performance of $78.8\% \pm 1.5$. Given the similarity of these results, and considering that the specific selected units are not exactly the same (as described in Section 3), we conclude that Experiment I reproduces previously reported findings for neuron-based representations. This result also provides external validation of our experimental protocol.

**Human performance is superior in the dictionary-based condition.**   Based on our main hypothesis that dictionary-based representations constitute a better basis for interpretability than neuron-based representations, we predicted that participants would perform better in the dictionary-based condition. Results

---

[5]Values inferred from Fig. 3 in Zimmermann et al. (2023)

across all three experiments are illustrated in Fig. 3. In Experiment I, the average performance across participants in the dictionary-based condition was • $83.5\% \pm 1.4$ [6] when compared to • $78.8\% \pm 1.5$ in the neuron-based condition. A Mann-Whitney U test revealed that participants performed significantly better in the dictionary-based condition $z = 3.12$, $p < .001$. This result was corroborated in both Experiments II (see examples of trials in Figs. A2–A4) and III, with a mean participant performance of • $85.1\% \pm 1.4$ *vs.* • $76.2\% \pm 1.6$, $z = 4.83$, $p < .001$ and a mean performance of • $81.5\% \pm 1.5$ *vs.* • $77.1\% \pm 1.6$, $z = 2.46$, $p = 0.01$, respectively. In subsequent analyses, we focus on the results from Experiments II and III.

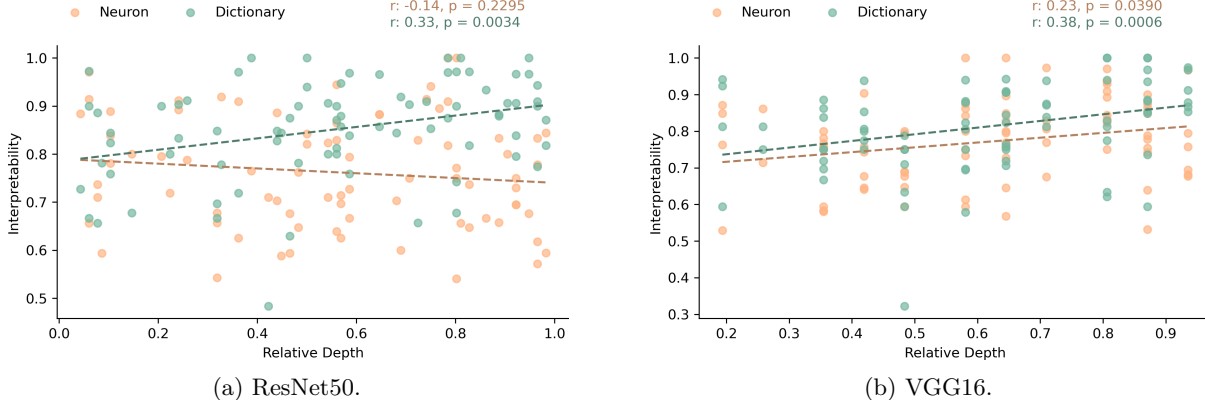

(a) ResNet50.  (b) VGG16.

Figure 4: **Correlation between human performance and the relative depth of the layer where the feature is extracted from for (a) ResNet50 and (b) VGG16.** For both models, we only find a significant correlation between interpretability and depth in the dictionary-based condition, after correcting for multiple comparisons (Bonferroni correction).

**The deeper the layer, the more prominent the benefits of dictionary-based representations.** Interestingly, our results suggest that the benefits of dictionary-based representations increase with relative layer depth in both models. Specifically, we observe a significant positive correlation between *interpretability* and *depth* for the dictionary-based condition only: Experiment I: $r = 0.35$, $p = 0.001$; Experiment II: $r = 0.33$, $p = 0.003$ for ResNet50 and $r = 0.38$, $p < 0.001$ for VGG16. In contrast, the correlations for the neuron-based condition are weaker and not statistically significant after Bonferroni correction: Experiment I: $r = 0.09$, $p = 0.4$; Experiment II: $r = -0.14$, $p = 0.23$ for ResNet50 and $r = 0.24$, $p = 0.04$ for VGG16. These patterns are illustrated in Fig. 4a and 4b, corresponding to Experiments II and III, respectively. To directly test this interaction, we fit a mixed-effects model on the pooled data from all three experiments: `Interpretability` $\sim$ `Representation` $\times$ `Depth` $\times$ `Model + (1|Subject) +` `(1|Experiment)`. The Experiment random effect was estimated at zero variance (boundary-singular fit, as expected with only three levels), and refitting without it yielded essentially identical fixed-effect estimates. A Type-III ANOVA confirms the predicted `Representation` $\times$ `Depth` interaction, $F(1, 15320.6) = 11.59$, $p < 0.001$. Post-hoc slope contrasts (*emtrends*), averaged over `Model`, show that the depth slope is significantly steeper for the dictionary basis than for the neuron basis (dictionary: $0.159$, 95% CI $[0.122, 0.195]$; neuron: $0.068$, 95% CI $[0.030, 0.105]$; contrast $= 0.091$, $z = 3.41$, $p < 0.001$). This directly confirms—without relying on the absence of a significant correlation—that the interpretability advantage of dictionary-based representations grows with layer depth. The three-way `Representation` $\times$ `Depth` $\times$ `Model` interaction did not reach significance ($F = 3.12$, $p = 0.078$), so we do not interpret the per-model slope differences as statistically significant, although the descriptive pattern (ResNet50 contrast: $0.138$, $p < 0.001$; VGG16 contrast: $0.044$, $p = 0.354$) is consistent with our results.

### 4.3 Comparison of model interpretability

While evaluating the interpretability of a specific machine learning model is important for validating its transparency for downstream use cases, it is equally important to understand what makes certain models

---

[6]The values reported correspond to a 95% confidence interval.

more interpretable than others. Doing so requires fair model comparisons. In this section, we report three main findings obtained by comparing ResNet50 and VGG16.

**ResNet50 exhibits lower axis-alignment than VGG16.** Superposition arises when a model attempts to represent more patterns than it has neurons, resulting in patterns that are encoded by populations of neurons rather than by single units. As the number of encoded patterns increases, individual dictionary elements are expected to be less aligned with any single neuron axis. As summarized in Fig. 5, we quantify the alignment of learned dictionary elements extracted with CRAFT with the neuron basis using Eq. 2. We find that ResNet50 exhibits lower average axis-alignment than VGG16 (0.40 *vs.* 0.45, $z = 6.01, p < .001$, Mann–Whitney U test), indicating that its dictionary elements are more distributed across neurons. This finding is consistent with a greater degree of representational sharing across neurons in ResNet50, a pattern commonly linked to superposition. Consequently, neuron-level analyses may obscure structure that only becomes visible in a dictionary-based basis, potentially affecting interpretations of model interpretability.

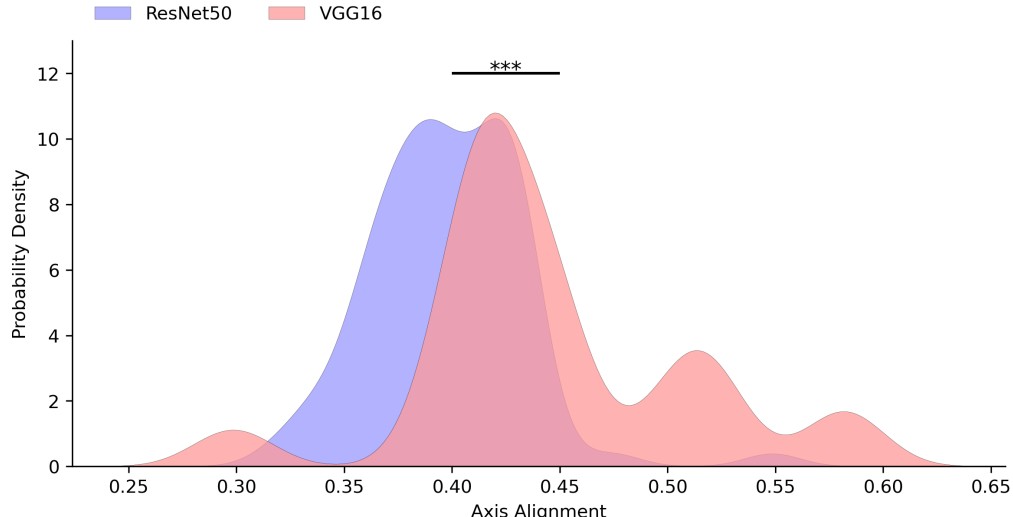

Figure 5: **Axis-alignment of dictionary elements.** We measure the axis-alignment of dictionary elements recovered by CRAFT for both models using the sparsity measure of Hoyer (2004) (see Eq. 2). Higher values indicate stronger alignment with individual neuron axes, *i.e.*, the recovered dictionary elements can be well approximated by single neurons, whereas lower values indicate more distributed representations across neurons. We find that ResNet50 exhibits significantly lower axis-alignment than VGG16: 0.40 *vs.* 0.45, $z = 6.01, p < .001$ (Mann Whitney U test). This difference is consistent with greater representational distributedness in ResNet50 a pattern commonly linked to superposition.

**ResNet50 is more interpretable than VGG16.** Within the Rashomon set of similarly accurate models— *e.g.*, ResNet50 achieves 76% accuracy compared to 72% for VGG16 on ImageNet, some models are expected to be more interpretable than others. A common assumption in mechanistic interpretability is that sparser representations should be more interpretable.

Based on the theoretical intuition that neural networks encode patterns in superposition and that interpretability improves when dictionary elements are sparse, we hypothesized that ResNet50 would exhibit greater interpretability than VGG16 in the dictionary-based condition. Our results corroborate this intuition: ResNet50 achieves significantly higher interpretability than VGG16 (85.1% ± 1.4 *vs.* 81.5% ± 1.5, $z = 2.14, p = 0.016$) in the dictionary-based condition. These findings contradict prior work (Zimmermann et al., 2024), which reported higher interpretability for VGG16 with 89.27% *vs.* 87.40% for ResNet50, underscoring the critical role of the representational basis when evaluating model interpretability.

**The choice of basis matters when comparing model interpretability.** Fig.3b depicts the average interpretability of ResNet50 and VGG16 for each representational basis (neuron-based vs. dictionary-based). While ResNet50 achieves higher interpretability than VGG16 in the dictionary-based condition, this advantage disappears when interpretability is assessed using neuron-based representations ($z = 0.56, p = 0.71$). These results suggest that a model's apparent interpretability critically depends on the choice of representational basis. Importantly, they challenge prior claims about model interpretability (Zimmermann et al., 2023), and highlight the need to consider dictionary-based representations when evaluating and comparing models.

## 5  Conclusion

In this work, we have investigated the suitability of neuron-based and dictionary-based representations as a basis for interpretability in two convolutional neural networks, ResNet50 and VGG16. Across three large-scale psychophysics experiments, we find converging evidence that the choice of representational basis significantly influences human interpretability: dictionary-based representations are consistently more interpretable than neuron-based representations, with the advantage increasing in deeper layers. Furthermore, we find models rely no less on dictionary elements than on neuron axes to make their decisions.

We also observe that the degree of representational distributedness in models, as measured by axis-alignment of dictionary elements, influences their apparent interpretability. ResNet50 exhibits lower axis-alignment than VGG16, indicating more distributed representations. This leads to comparable interpretability when assessed using neuron-based representations but superior interpretability when evaluated using dictionary-based representations. This finding is particularly notable given that ResNet50 is the model with higher classification accuracy.

These findings demonstrate that the choice of representational basis matters when comparing models, suggesting a need to reinterpret prior results. Overall, our results highlight that dictionary-based representations not only constitute a superior basis for interpretability but also that comparing models using neuron-based representations alone can lead to misleading conclusions.

**Limitations and future work.** Our study is not without limitations. First, the methodology proposed by Borowski et al. (2021) and followed by Zimmermann et al. (2023) utilizes the entire ImageNet validation set with the goal of studying a broad range of stimuli and thereby increasing the likelihood of identifying stimuli that are representative of a neuron's selectivity. While this motivation is sound, the trade-off is that neurons selective to class-specific patterns (*e.g.*, fish scales) will be maximally activated by stimuli from the corresponding classes (*e.g.*, fish) and minimally activated by stimuli from other classes (*e.g.*, dogs). In such cases, the task can be solved trivially using semantics. The design of Experiment II reflects an initial attempt to mitigate this confound. However, manual inspection of the trials by the authors suggests only partial success in addressing this challenge as there were residual confounds: even after matching semantic labels, low-level statistics, such as dominant color or average texture, can correlate with class identity and serve as shortcuts. Additionally, the WordNet hierarchy expansion occasionally matched semantically related but non-identical categories. Fully eliminating semantic cues likely requires richer image metadata or procedural controls beyond label matching. We leave further refinement of the experimental protocol to future research.

Second, this work focused on investigating whether analyzing individual neurons reflects a mixture of pattern detectors in superposition. This approach enabled a fair one-to-one comparison of elements recovered from both neuron-based and dictionary-based representations, providing evidence that dictionary learning methods can improve our understanding of model representations. However, future research should examine the interpretability of dictionary elements recovered by applying dictionary learning to full layer activations.

Finally, our experiments were limited to two models. While two models were enough to illustrate that the choice of representational basis influences interpretability comparisons, future work should expand the range of architectures to understand why certain models are inherently more interpretable than others.

## Acknowledgement

This work was funded by the ONR grant (N00014-24-1-2026), NSF grant (IIS-2402875), the ANR-3IA Artificial and Natural Intelligence Toulouse Institute (ANR-19-PI3A-0004), the Regional Government of Valencia in Spain (Resolución de la Conselleria de Industria, Turismo, Innovación y Comercio, Dirección General de Innovación) and the European Union's Horizon Europe research and innovation program (ELIAS; grant agreement 101120237). The computing hardware was supported in part by NIH Office of the Director grant S10OD025181 via the Center for Computation and Visualization (CCV) at Brown University. J.C. and N.O. are supported by Intel Corporation and J.C. is also supported by a grant by Banco Sabadell Foundation. T.F is supported by the Kempner Institute Research Fellowship.

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

# A   Selection of dictionary elements

Our hypothesis is that applying dictionary learning to neuron-based representations enables the recovery of directions corresponding to multiple patterns encoded in superposition, yielding representations that are more interpretable than individual neurons. To evaluate this hypothesis, for each neuron in the neuron-based condition, we aim to recover the underlying patterns it encodes by learning a complementary dictionary over neuron activations.

Concretely, following recommendations of Fel et al. (2023b), we learn a dictionary with $k = 10$ elements. This choice of $k$ is intended to be sufficiently large to capture the meaningful patterns encoded by a neuron while satisfying $k > s$, where $s$ denotes the (unknown) number of latent patterns encoded in superposition. Unused dictionary elements are expected to capture residual noise.

From the learned dictionary, we select a single element to compare directly with the original neuron. A key methodological question is how to select this dictionary element. We consider three strategies: (a) random selection; (b) selection based on predictive importance; and (c) selection based on relevance, measured by activation frequency. We adopt strategy (c), as it introduces the least amount of bias into the comparison.

Random selection (a) is undesirable because, unless the model is in an extreme superposition regime, any given neuron is expected to encode at most $s < k$ different patterns in superposition. Consequently, randomly selecting a dictionary element could lead to an element that captures noise rather than a meaningful pattern, which would unfairly disadvantage dictionary-based representations. On the other hand, selecting elements based on their predictive importance (b) would bias the comparison in favor of dictionary-based representations, since the neurons themselves were not selected according to their importance for the model's predictions. In contrast, selecting the dictionary element with the highest activation frequency (strategy (c)) provides a principled and symmetric criterion that reflects how consistently a pattern is expressed in the data, without privileging either representation.

## B  Experimental confounds

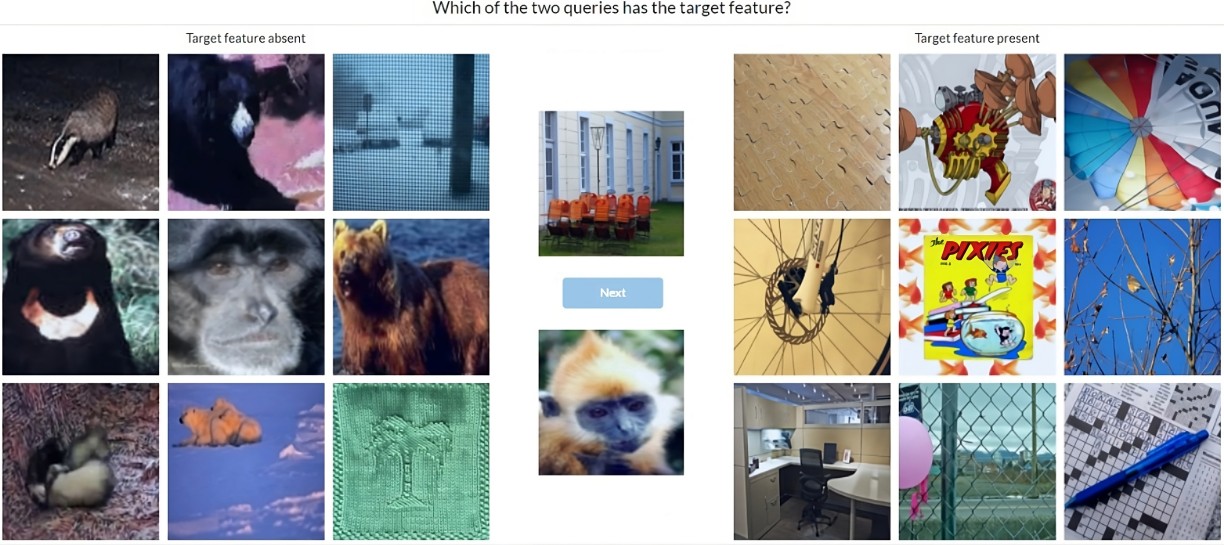

Figure A1: **Illustration of the role of semantics**. Example of a trial from Experiment I in the neuron-based representation condition. In this case, the task can be trivially solved by relying on semantic grouping. By observation of the minimally activating stimuli (left panel), it is easy to conclude that the neuron of interest is not a monkey detector, yet, it is hard to articulate what visual pattern is captured by the neuron (images in the right panel).

## C   Examples of trials from Experiment II

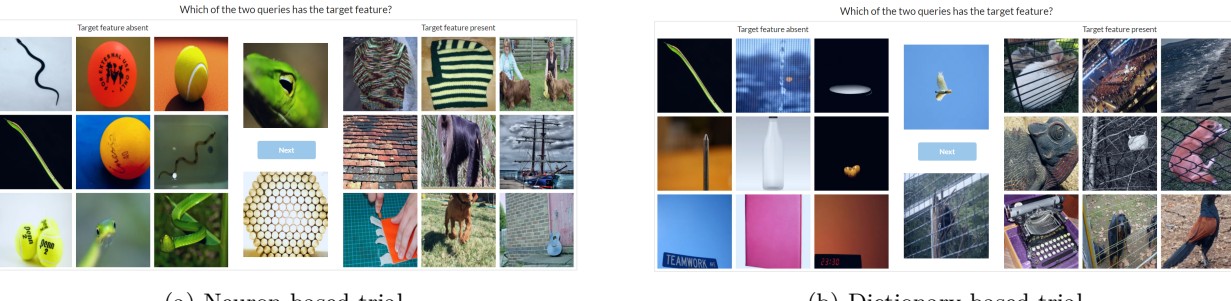

(a) Neuron-based trial        (b) Dictionary-based trial

Figure A2: **Layer2.** This figure illustrates a trial used to assess the features encoded in layer2.0 either by the neuron 52 (a) or at least partially through the neuron 52 (b).

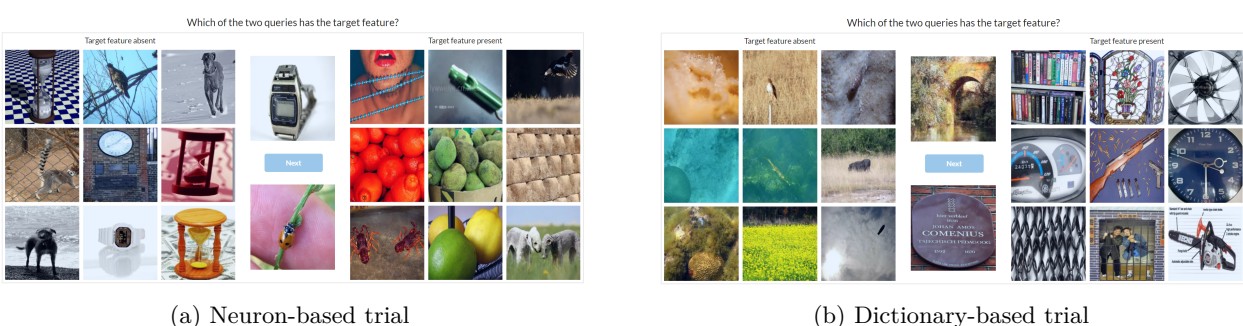

(a) Neuron-based trial        (b) Dictionary-based trial

Figure A3: **Layer3.** This figure illustrates a trial used to assess the features encoded in layer3.1 either by the neuron 957 (a) or at least partially through the neuron 957 (b).

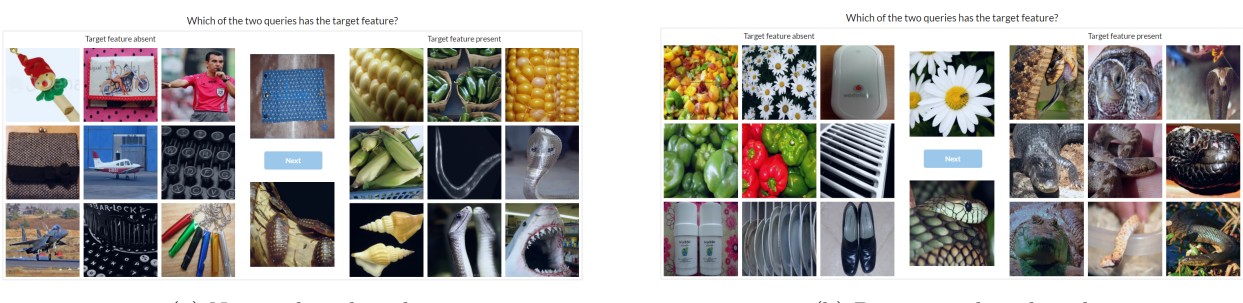

(a) Neuron-based trial        (b) Dictionary-based trial

Figure A4: **Layer4.** This figure illustrates a trial used to assess the features encoded in layer4.2 either by the neuron 259 (a) or at least partially through the neuron 259 (b).

## D    Sensitivity analysis

To complement the sample-size prior from Zimmermann et al. (2023), we performed a sensitivity analysis based on our retained sample sizes. For the main dictionary-vs-neuron comparisons, each experiment can be approximated as a two-group comparison with approximately 65 participants per condition. With $\alpha = 0.05$ and 80% power, this design is sensitive to medium-sized effects: Cohen's $d = 0.49$ or larger. The observed dictionary-vs-neuron effects were d = 0.56, d = 0.94, and d = 0.45 for Experiments I–III, respectively. Thus, our individual experiments were sensitive to medium-sized dictionary-vs-neuron differences, and the observed effects were generally of this magnitude. For the analyses pooling across experiments, we rely on the mixed-effects models reported in the main text, which account for the structure of the data by including participant-level variability and experiment/model structure.

