# OpenReview forum: "Choosing the right basis for interpretability: Psychophysical comparison between neuron-based and \\ dictionary-based representations"
_TMLR — Under review for TMLR_

### Review · Reviewer_88n4 · 2026-04-08

**Summary Of Contributions:**

The paper compares how visually-coherent (ie, interpretable) neuron- and dictionary-based representations are to human participants. Specifically, participants are asked to judge which of two test images is similar to a set of maximally activated images based on the two representations. The paper finds that, across two models (VGG and ResNet), humans are consistently better at performing the task when presented with images that are maximally activated  based on dictionary representations compared to neuron-based representations. The paper also finds that model interpretability (operationalized as matching accuracy in the task) based on dictionary representations aligns with model performance, while neuron-based interpretability doesn’t differ between the two models. Based on these findings, the authors conclude that dictionary-based representations are superior for interpretability.

Strengths:

1. The paper tackles an important and timely question of which representations are more interpretable — dictionary- or neuron-based ones.
2. The experiments are large-scale (for human experiments) and are rigorous (the authors perform a power analysis to determine the number of participants based on similar prior work; implement catch trials and filtering for participants who took too long/short to complete the experiment;  find and eliminate  a potential confound in previous work).
3. The paper is well-written and clearly laid out.


Weaknesses:
The major weakness is that based on the findings, I’m not convinced that dictionary-based representations provide a better basis for interpretability than neuron-level analyses. The dictionary learning procedure used in the current work involves substantial filtering, which isn’t applied to neurons, which may encourage more coherence in the candidate images. Thus,  it’s possible that the advantage of dictionary-representations observed in human experiments isn’t due to the representation per se but rather to the filtering procedure.

Some smaller points:

1. The paper considers two models, both of which are CNNs. I would have liked to see a vision transformer since they’re very widely used.
2. The claims about superposition based on axis-alignment are bit too strong.

**Audience:**

Yes

**Audience Explanation:**

The question of whether neuron- or dictionary-based representations provide a closer alignment with human judgements is of great interest to the field of interpretability.

**Claims And Evidence:**

No

**Claims Explanation:**

Based on the findings, I’m not convinced that dictionary-based representations provide a better basis for interpretability than neuron-level analyses. The dictionary learning procedure used in the current work involves substantial filtering, which isn’t applied to neurons, which may encourage more coherence in the candidate images. Thus, it’s possible that the advantage of dictionary-representations observed in human experiments isn’t due to the representation per se but rather to the filtering procedure.

**Requested Changes:**

Critical:

1. To make a claim that dictionary representations are more interpretable than neuron-based ones, it is critical to disentangle representation from filtering. Here’s what I mean. The starting point are the 80 neurons from Zimmermann et al. (2023) for ResNet and 80 random neurons for VGG. For the neuron-condition, the images presented to participants come from 150 top- and 20 bottom-activated images for each neuron. For the dictionary-condition, the procedure is different: take 300 max-activated images for the target neuron and apply CRAFT to learn 10 dictionary elements over the layer that contains the target neuron. From the 10 dictionary elements, select the one that is maximally activated for the 300 images. The images are then re-ranked according to their activations along the top dictionary element. Thus, dictionary representations seem to have 2 advantages over neuron-based ones: a) the selection process would favor coherence, which is the variable tested in the human experiments; b) dictionary representations use the information from the entire layer.

I think some controls are necessary to really pin down the advantage of dictionary-based representations (especially since the differences in human performance for the two representations aren’t whopping). What if one had some selection procedure for neurons in a given layer — eg, what if one looked at the neuron that’s most activated for the 300 images in a layer or took top-10 most activated neurons in a layer and out of those picked the neuron with the lowest entropy of activations for the 300 images? What if one removed the selection procedure for dictionary elements — eg,  learned k=1 dictionary elements or picked a random one? The authors do argue in the Appendix that picking a random element would put the dictionary-based approach in a disadvantage. But it appears that there’s a disadvantage for neuron-based representations in the current approach. Without some of these comparisons, it’s really hard to claim that the difference in human performance between the dictionary- and neuron-based approaches is due to the representation and not the selection procedure. I think it’s an important distinction to draw since it would change the recommendation to practitioners.

I understand that running additional human experiments may not be feasible. I wonder if one could replicate the current human results with some ML metrics (e.g., pairwise cosine over CLIP embeddings, etc) or have a strong vision model complete the task. And if so, use these metrics/model to quickly test some of the hypotheses I mentioned above.

2. The patterns and magnitude of the correlations reported in Fig. 4 don’t always correspond to their description in text. What’s going on there?

Nice-to-have’s:

1. Could you explain how the 80 neurons for ResNet were chosen? The paper points to Zimmermann et. al with no explanation but this should be explained in text rather than require a reader to look up the reference.
2.  Could you comment on this sentence in the Conclusion: “However, manual exploration of the trials by the authors suggests only partial success in addressing this challenge.“ — i.e., what do you mean by partial and what are the remaining issues? The description of Exp. 2 sounded like the semantic confound was addressed so I was a bit confused reading this part in the Conclusion.
3. I’d suggest toning down the claims about superposition based on axis-alignment. Axis-alignment is a proxy for superposition and the paper doesn’t test superposition directly. I think it’s worth stating this more explicitly and being clear that the paper provides the evidence consistent with/suggestive of superposition but doesn’t test it directly.
4. More generally, could you speak about how you see your findings being used for downstream tasks?

---

> ### Author Response · Authors · 2026-04-30
> **Response to reviewer 88n4 (1)**
>
> We thank the reviewer for their thoughtful and constructive feedback. We address each point below.
>
> > Major
>
> ## Concern about the fairness of the distributed condition
> We understand the reviewer's concern and appreciate the chance to address it. We would like to reframe the comparison in a way that we think speaks more directly to the question of fairness.
>
> The implicit assumption behind the concern is that neurons are the correct atomic unit of explanation, and that the dictionary procedure introduces an additional filtering step that the neuron condition does not benefit from. We think this is exactly the assumption the experiment is designed to test. If neurons were genuinely the correct atom, then any post-hoc filtering applied within the neighborhood of a neuron should have only a marginal effect on interpretability as there would be little additional structure for the filtering to exploit. The fact that the filtering yields a substantially more interpretable direction is itself informative about whether the neuron is the right unit. Under the neuron-as-atom view, the comparison is not unfair: it is a test that view is expected to pass.
>
> This connects to the broader design choice, which we should make explicit. To run the test described above, we need a one-to-one mapping between each neuron and a distributed direction, so that the two conditions probe the same underlying unit and only differ in the basis used to describe it. This is precisely why we study superposition locally rather than globally. Methods like sparse autoencoders study superposition globally: they are trained on activations elicited by a large, diverse dataset that spans the full activation space of a layer, and they learn an overcomplete dictionary covering that entire space, obtaining a set of directions for the layer as a whole. We instead restrict attention to a small region of activation space, namely the region populated by the top-300 images for a given neuron. Although these images elicit activity across all dimensions of the layer, they share the property of strongly activating the target neuron, so they probe a particular sub-region of the layer's activation space. Within that sub-region, CRAFT recovers k=10 directions, and we retain the element best aligned with the local region. This yields the one-to-one neuron-to-direction mapping the comparison requires, and lets us ask: in the neighborhood of a given neuron, can the layer's local geometry be decomposed into directions that are more interpretable than the neuron itself?
>
> That said, we want to engage with the reviewer's concern directly. To target the right control experiments, could the reviewer clarify what they mean by:
> (a) "the selection process would favor coherence"
> (b) "dictionary representations use the information from the entire layer" — why is using layer-wide information considered a limitation?
>
> ## Discrepancy in text vs figure
> Thank you for flagging this — there is indeed a discrepancy we missed. We will update the text to correctly reflect the numbers reported in Figure 4.
>
> ## Transformers
> We share the reviewer's interest in vision transformers. We restrict the present study to CNNs for two reasons: NMF requires non-negative inputs, which CNN post-ReLU activations satisfy by construction but transformer activations do not, and CNNs are sufficient to test our central hypothesis about neuron-vs-dictionary interpretability. Extending the framework to transformers,  likely with a decomposition method appropriate for signed activations, is a natural next step that we will note in the discussion.
>
> ## Explanation of neuron selection
> We will add a brief description of the selection procedure to the methods rather than only referencing Zimmermann et al. (2023). For completeness, the procedure is: "For each of the investigated models, we randomly select 84 units […] by first drawing a network layer from a uniform distribution over the layers of interest and then selecting a unit, again at random, from the chosen layer. This scheme is used instead of randomly drawing units from a uniform distribution over all units since CNNs typically have more units in later layers."
>
> ## "Partial success" in addressing the semantic confound
> We apologize for the confusion. Experiment 2 addresses the semantic confound, i.e., cases where matching is driven by category-level information rather than feature-level coherence. However, manual inspection of the trials revealed residual low-level visual confounds that are not category-driven: e.g., in some trials, the stimuli unrelated to the target feature happened to be predominantly yellow, allowing matching on a basis other than the intended feature. We will rewrite the relevant passage to clarify.

---

> > ### Author Response · Authors · 2026-04-30
> > **Response to reviewer 88n4 (2)**
> >
> > ## Superposition claims
> > We agree with the reviewer. Axis-alignment is a proxy for superposition rather than a direct test of it, and our measure speaks more directly to distributedness than to sparsity per se. We will revise the relevant section to (i) discuss distributedness as the measured quantity, (ii) flag the link to sparsity as an assumption, and (iii) frame our results as consistent with — rather than a direct test of — superposition.
> >
> > ## Downstream impact
> > The primary contribution of this work is to provide a human-centered account of an ongoing debate in interpretability: whether the meaningful units of analysis are individual neurons or distributed directions in the activation space. Existing arguments in this debate have largely been theoretical or based on automated metrics, but interpretability is ultimately for humans, and our results provide direct human evidence on the question.

---

### Review · Reviewer_67sh · 2026-04-22

**Summary Of Contributions:**

When seeking explanations for neural network decisions, common approaches are often neuron-based, meaning they treat individual neurons as the unit of explanation. But the most meaningful or interpretable directions may not be neuron-aligned. This paper measures the human interpretability of neuron-based vs dictionary-based representational bases. Dictionary-based representations are learned via CRAFT, a non-negative matrix factorization technique. The authors operationalize interpretability via visual coherence: the extent that humans can reliably identify the common visual pattern in images that maximally activate a direction and generalize that pattern to new images. Across three human experiments, the authors find that, on average, dictionary-based directions are more human interpretable than neuron-aligned ones. They also report a relationship with layer depth where interpretability increases with depth for dictionary- but not neuron-based representations. In comparisons between ResNet50 and VGG16 the authors find that the choice of basis matters for evaluating interpretability: ResNet50 is more interpretable than VGG16 when using dictionary-based representations but this pattern does not occur with neuron-based representations.


Strengths:
- The research question is well posed
- The human experiments seem well designed and build on established paradigms


Weaknesses:
- There are some issues with the stats, but they are addressable.
- Clarity could be improved, particularly concerning the experimental design and the discussion of representational properties like superposition and sparsity.
- Only two models investigated

**Audience:**

Yes

**Audience Explanation:**

This work will be of interest to those interested in explainable/interpretable deep learning. There is also overlap with the computational neuroscience community in the discussion of different representational properties and methods for finding meaningful directions in high-dimensional neural spaces.

**Claims And Evidence:**

No

**Claims Explanation:**

There are currently some issues with how statistical hypothesis tests are employed. But once these are addressed, I expect the updated claims to be well-supported. The data collected appears to be appropriate to answer the question.

**Requested Changes:**

# Critical/Major
## 1. Statistical hypothesis testing
Throughout the results section, you analyze Interpretability as a function of three independent variables: Representation (neuron vs dictionary), Model (ResNet50 vs VGG16), and Depth (integer scale). In the section "The deeper the layer, the more prominent the benefits of dictionary-based representations" you are describing essentially an interaction: "we observe a significant positive correlation between interpretability and depth for the dictionary-based condition only". The absence of a significant correlation is not evidence of absence (at least in frequentist stats). It would be better to test for this interaction directly in a multiple regression model, and then to compare the slopes in posthoc tests. This is slightly complicated by the fact that Model and Representation are between-subject variables while Depth is within-subject. To account for this, I would recommend a mixed-effects model with a random intercept for subject:

Interpretability ~ Representation * Depth * Model + (1 | Subject).

This will also account for the fact that data collection was split over three experiments. Based on Figure 4, you will likely find a Representation * Depth interaction and then you can test that the slope is significantly greater for dictionary compared to neuron-based representations, averaged over model. This would lead to a very similar interpretation to what is currently in the manuscript, but quantified explicitly as an interaction. If instead you find a 3-way interaction Representation X Depth X Model, then you will end up looking at the four different slopes currently depicted in Figure 4 and do a interaction contrast to test how the difference between dictionary and neuron based representation is different between models. This may lead to a slightly different interpretation, depending on the results.

I think it is fine to first analyze each experiment separately showing that the dictionary-based representations are more interpretable than the neuron based ones and that you replicated this across three separate experiments. But as soon as you start comparing models and pooling across experiments, it is not sufficient to report the absence of a significant difference in once case and the presence in another. You should fit a linear model to look for these interactions, then, only if significant, perform the posthoc tests.

## 2. Clarity
It is difficult to ascertain the exact experimental design employed and what exactly each participant did, partly because this information is spread over several sections. You write,
"Each experiment consists of 1,600 trials (80 units ×2 conditions × 10 trials per unit)"
"participants performed 40 trials of the same task, with each trial corresponding to a different unit (a specific neuron axis or a dictionary element, depending on the condition)"
How exactly were units assigned to participants? Is it the case that all participants experienced units from all depths? When you say "10 trials per unit" and "each trial corresponding to a different unit", does that mean exactly the same trial (same images) were presented to 10 different subjects?

# Optional/Minor

## 3. Power analysis
You describe that a previously reported power analysis revealed how many participants would be required to "obtain statistically robust results". A power analysis reveals how many participants are required to reliably detect an effect of a certain size. Past research should tell you what effect size is expected so your power anaylsis should tell you how many participants you need to detect that effect size. You should perform your own power analysis based on your particular experimental design, especially taking into account the fact that in some of your analyses you are pooling across "Experiments".
## 4. Relevant work
You may be interested in some relevant work that sits at the intersection of computational cognitive neuroscience, e.g.

	- Mahner, Florian P., et al. "Dimensions that matter: Interpretable object dimensions in humans and deep neural networks." (CCN2023).
	- Journal article from the same authors https://www.nature.com/articles/s42256-025-01041-7
	- Morcos, Ari S., et al. "On the importance of single directions for generalization." arXiv preprint arXiv:1803.06959 (2018)
## 5. "Psychophysics"
The term "psychophysics" seems like a bit of a misnomer in this context. "Psychophysics" evokes a setting where experimenters systematically vary some physical property of the stimulus and observe the relationship between that property and perception. "Behavioural experiment" seems more appropriate here but this is very minor so feel to disregard.
## 6. Appendix missing
Appendix not included? reference to figure A1 but no such figure
## 7. Axis-alignment as a proxy for superposition
I find the logic in section "Axis-alignment as a proxy for superposition" somewhat shaky.
In this work, superposition is equated with less axis-alignment which is measured as sparsity. When we talk about superposition increasing with the number of patterns to be encoded, we are really talking about the orthogonality of the representation. As we move beyond N patterns, the representations will need to be correlated. This is not the same as sparsity. A one-hot representation is maximally orthogonal and maximally sparse. A random rotation of such a representation will still be orthogonal, but it will be no longer sparse. Of course, the reverse is not true---sparsity does constrain but does not determine orthogonality. Why networks learn the particular representations they do is still somewhat an open question. They learn representations that are more class selective than would be optimal (https://arxiv.org/abs/2003.01262) and seem to have an implicit bias towards shared representations that results from the learning dynamics (https://proceedings.mlr.press/v162/saxe22a.html). It seems like an over simplification to equate superposition with axis-alignment with sparsity with the "number of sparse features" encoded.

---

> ### Author Response · Authors · 2026-04-30
> **Response to Reviewer 67sh**
>
> We thank the reviewer for their detailed and constructive feedback. We address each point below.
>
> > Critical/Major
>
> ## Statistical hypothesis testing.
> We agree with the reviewer, and we are currently working on the more sophisticated statistical analysis aligned with the reviewer’s suggestion. We will update this rebuttal with the results shortly.
>
>
> ## Clarity
> To clarify: the same images were not shown to 10 different participants. The design is as follows. Within a given (model, condition) pair, we select 80 units distributed across layers. For each unit, we construct 10 different trials, each using a different set of stimuli to illustrate that unit. For each participant, we then (i) randomly sample 40 of the 80 units, and (ii) for each sampled unit, randomly draw 1 of its 10 trials. Each participant therefore sees 40 trials, each corresponding to a different unit, and never sees the same trial twice. We will consolidate this description into a single subsection of the methods to make the design unambiguous.
>
> > Optional/Minor
>
> ## Power analysis.
> We acknowledge this limitation. The Zimmermann et al. analysis was used as a prior given the closely matched experimental design (same task, same image source, similar unit pool). In the revision, we will supplement this with a sensitivity analysis based on the observed effect sizes from our own data.
>
> ## Related Work
> Thank you for these pointers!
>
> ## "Psychophysics."
> Thank you for the suggestion. We respectfully retain the term psychophysics. Modern usage of the term encompasses rigorous behavioral paradigms that quantitatively measure perception, not only classical threshold/detection paradigms. Borowski et al. (2021), on whose protocol we build, also use psychophysics to describe this exact task type. We will add a brief footnote acknowledging the broader usage.
>
> ## Appendix
> Apologies for the confusion — the appendix was submitted as a separate file. We will ensure it is properly linked in the revision.
>
> ## Axis-alignment as a proxy for superposition.
> This is a fair critique, and we will revise accordingly. The reviewer is correct that axis-alignment is a measure of distributedness rather than sparsity, and that orthogonality and sparsity are different properties (a rotated one-hot code being the canonical counterexample). Our intended argument was indirect: we assume that more distributed representations carry more features, and that the features in such a representation are correspondingly sparser, which would link distributedness to sparser models. We will revise this section to (i) discuss distributedness directly as the measured quantity, (ii) flag the sparsity step as an assumption rather than an implication, and (iii) tone down the equation of superposition with axis-alignment.

---

> > ### Author Response · Authors · 2026-05-07
> > **Additional statistical analysis**
> >
> > We have now run the more sophisticated statistical analysis aligned with the reviewer's suggestion and we detail our findings below. We fit the recommended mixed-effects model on the pooled data from all three experiments:
> > > Interpretability ~ Representation * Depth * Model + (1 | Subject) + (1 | Experiment)
> >
> > The Experiment random effect was estimated at zero variance (boundary-singular fit, as expected with only three levels); refitting without it yields essentially identical fixed-effect estimates.
> > The Type-III ANOVA confirms the predicted Representation × Depth interaction, $F(1, 15320.6) = 11.59, p < 0.001$. The post-hoc slope contrast (emtrends), averaged over Model, shows that the depth slope is significantly steeper for the dictionary basis than for the neuron basis (dictionary: 0.159, 95% CI [0.122, 0.195]; neuron: 0.068, 95% CI [0.030, 0.105]; contrast = $0.091, z = 3.41, p < 0.001$). This is the stronger test the reviewer suggested, and it confirms — without relying on the absence of a significant correlation — that the interpretability advantage of dictionary-based representations grows with layer depth. The three-way Representation × Depth × Model interaction did not reach significance ($F = 3.12, p = 0.078$), so we do not interpret the per-model slope differences as statistically significant, although the descriptive pattern (ResNet50 contrast: $0.138, p < 0.001$; VGG16 contrast: $0.044, p = 0.354$)  is consistent with our previous results.

---

> > ### Comment · Reviewer_67sh · 2026-05-26
> >
> > Thanks for your response. The discussion responses are satisfactory in principle, but I was expecting a revised manuscript to be uploaded. My recommendation assumes the promised revisions are incorporated into the manuscript.

---

> > > ### Author Response · Authors · 2026-06-06
> > > **Revised manuscript**
> > >
> > > As per your request, we have uploaded a revised version of the manuscript. The implemented changes are marked with blue font. We hope to have addressed all your concerns.
> > >
> > > Thank you again for the detailed and constructive feedback.
> > >
> > > Cheers
> > >
> > > Authors

---

> > > > ### Comment · Reviewer_67sh · 2026-06-11
> > > >
> > > > Thanks for the updated manuscript. The revision is much clearer, more convincing, and more precise. All of my critical requested changes have been addressed.

---

### Review · Reviewer_wtPe · 2026-04-23

**Summary Of Contributions:**

This paper runs experiments with humans to measure interpretability of differently defined model units across different models. They show that humans are better able to match images to a model unit's preferred features when those units are defined based on a NMF-derived basis. This is truer at later layer in the models, and ResNet has higher absolute interpretability.

**Audience:**

Yes

**Audience Explanation:**

The benefit of using different bases to describe layer activation is a major interest area in interpretability. This study contributes empirical work with humans, which is somewhat rare in this field.

**Claims And Evidence:**

Yes

**Claims Explanation:**

The stated claims about the outcomes of the experiments are in line with the results. I do have questions about the interpretation and connection to the larger literature however (see below).

**Requested Changes:**

I mainly have clarifications I would like to see, to make the methods and discussion clearer.

1. Are the neurons in convolutional layers actually individual units? Or are they global averages of channels/'feature maps'?
2. For the dictionary creation, what images are used? Is it just the 300 most activating for the comparison neuron? Or the full set of images? If it is just the 300, I would be concerned that that introduces some bias in the dictionary learning. In either case, the text could indicate this more clearly.
3. For dictionary creation, what neurons are used? The methods say "We then applied CRAFT to learn a dictionary of k=10 elements over the activations of that neuron’s layer" but the discussion says "future research should examine the interpretability of dictionary elements recovered by applying dictionary learning to full layer activations." Were the dictionaries here not already made with full layer activations?
4. The choice of k=10 elements is confusing. Do the authors think there are only 10 latent visual features that these layers are trying to represent? Also the introduction and discussion frame this work as relating to superposition but superposition assumes there are more features than neurons. It seems instead the work here is more about dimensionality reduction, as k is less than the number of neurons.
5. The authors mention how sparse models are assumed to be more interpretable and say that it therefore makes sense that ResNet dictionaries are more interpretable than VGG16, however the sparsity of these dictionary representations were not measured (the axis alignment is not a measure of the sparsity of the dictionary response as I understand it).
6. Is the stronger difference between neuron and dictionary interpretability at later layers of resnet present in Experiment 1? It seems like the neuron interpretability could be impacted by the semantic constraints imposed in Experiment 2. In networks trained to do object classification, the existence of neurons tuned to semantic categories in the later layers makes sense and I think controlling for that as is done in experiment 2 is a double edged sword.

---

> ### Author Response · Authors · 2026-04-30
> **Response to Reviewer wtPe**
>
> We thank the reviewer for their careful reading and for recognizing the value of contributing human data to interpretability research. We address each point below.
>
>
>
>
> ## Clarification for the usage of neuron
> This is a fair question. Following common usage, we use "neuron" as a general term for the unit of a layer: an individual neuron in fully-connected or ReLU layers, and a channel (feature map) in convolutional layers. We will make this convention explicit in the revised text.
>
> ## Dictionary creation — which images and which neurons?
> CRAFT is applied to the N=300 maximally activating images for the target neuron, not to the full ImageNet validation set. This follows the procedure recommended in Fel et al. (2023), where N is chosen to yield a sufficient activation matrix for NMF while keeping the procedure tractable. The current text states that "we selected its top N=300 maximally activating images … We then applied CRAFT to learn a dictionary of k=10 elements over the activations of that neuron's layer," and we agree that it can be read ambiguously. In the revised version of the manuscript, we will include a figure to make the pipeline unambiguous.
> The discussion sentence the reviewer quotes was intended to point toward future work using full-dataset activations (rather than the top-300 subset), not full-layer activations. We will correct the wording, since the dictionaries in the present study are already learned over the full layer's channel activations for those 300 images.
>
> ## Why k=10, and how does this relate to superposition vs. dimensionality reduction?
> This is an important point and we appreciate the chance to clarify our framing.
> We do not claim that a layer represents only 10 latent features. Rather, our view is that superposition can be studied locally as well as globally, and k=10 reflects that local perspective.
> To unpack the distinction: methods like sparse autoencoders (SAEs) study superposition globally. They are trained on activations elicited by a large, diverse dataset that spans the full activation space of a layer, and they learn an overcomplete dictionary that covers that entire space. In contrast, we restrict attention to a small region of the activation space, namely the region populated by the top-300 images for a given neuron. Although these images elicit activity across all dimensions of the layer, they share the property of strongly activating the target neuron, so they probe a particular sub-region of the layer's activation space. Within that sub-region, CRAFT recovers k=10 directions.
> Under this local view, we interpret the procedure as dimensional expansion in the same spirit as an SAE: we are looking for more feature directions than the single neuron axis we started from, in order to disentangle what may be superposed in the neighborhood of that neuron. Viewed globally, k=10 is smaller than the layer width K, so the procedure also has the surface form of dimensionality reduction. The two readings are not in conflict so much as they reflect whether one looks at the operation from the perspective of the full layer or the perspective of a single neuron's neighborhood.
> We will revise the introduction and discussion to make this local-vs-global framing explicit and to avoid implying that we treat k=10 as the true number of features in the layer.
>
> ## Sparsity vs. axis-alignment.
> The reviewer is correct that axis-alignment is not itself a measure of sparsity, but a measure of how distributed the representation is. Our reasoning was indirect: we assume that more distributed representations carry more features, and that each feature in such a representation is correspondingly sparser. Hence, the link to sparser models. Since this chain rests on an assumption rather than a measurement, we will revise the manuscript to discuss distributedness directly and to flag the sparsity step as an assumption rather than an established property of the dictionaries.
>
> ## Is the neuron-vs-dictionary gap at later ResNet layers also present in Experiment 1?
> Yes. The same pattern holds in Experiment 1: the correlation between layer depth and interpretability is r = 0.35 (p = 0.001) in the distributed (dictionary) condition, versus r = 0.09 (p = 0.4) in the neuron condition. We will add this analysis to the revised manuscript, since it addresses exactly the reviewer’s concern that the Experiment-2 semantic constraints might be driving the effect. Thank you for the suggestion to carry out this analysis.

---

> > ### Comment · Reviewer_wtPe · 2026-05-26
> >
> > Thanks for the response. I think the planned clarifications and additional analysis will be very helpful for readers.

---

> > > ### Author Response · Authors · 2026-06-06
> > > **thank you!**
> > >
> > > Thanks so much! We have uploaded a revised version of the manuscript for your review.

---

### Author Response · Authors · 2026-06-06
**Updated version of the paper uploaded as a new PDF**

Dear Reviewers,

Thank you again for your detailed and constructive feedback. We have carefully reviewed all your comments and incorporated them in a revised version of the manuscript, which we have uploaded as a new PDF. The changes are highlighted in blue font. We truly hope to have addressed all your concerns and look forward to hearing from you.

Best wishes

Authors